# Targeted DNA Sequencing of Cutaneous Melanoma Identifies Prognostic and Predictive Alterations

**DOI:** 10.3390/cancers16071347

**Published:** 2024-03-29

**Authors:** Alexandra M. Haugh, Robert C. Osorio, Rony A. Francois, Michael E. Tawil, Katy K. Tsai, Michael Tetzlaff, Adil Daud, Harish N. Vasudevan

**Affiliations:** 1Department of Medicine, Division of Hematology/Oncology, Helen Diller Comprehensive Cancer Center, University of California San Francisco, San Francisco, CA 94142, USA; alexandra.haugh@ucsf.edu (A.M.H.); katy.tsai@ucsf.edu (K.K.T.); adil.daud@ucsf.edu (A.D.); 2Department of Radiation Oncology, University of California San Francisco, San Francisco, CA 94143, USAmichael.tawil@ucsf.edu (M.E.T.); 3Department of Neurological Surgery, University of California San Francisco, San Francisco, CA 94143, USA; 4Department of Dermatology, University of California San Francisco, San Francisco, CA 94143, USA; 5Department of Pathology, University of California San Francisco, San Francisco, CA 94143, USA

**Keywords:** melanoma, MAPK pathway, BRAF, NRAS, NF1, triple wild type, immune checkpoint inhibitors, biomarkers

## Abstract

**Simple Summary:**

Melanoma is the most lethal form of skin cancer and has seen a rising incidence globally over the last 10 years. A majority of melanomas arise from the skin as a result of exposure to ultraviolet radiation. However, even within this group of “cutaneous melanomas”, there is considerable clinical and molecular heterogeneity. In this study, we sought to understand how different genetic “drivers” in a group of 254 patients with cutaneous melanomas influenced overall survival and response to immunotherapy. We found that *NRAS* mutation correlated with decreased survival and that tumor mutational burden (an estimate of the amount of genetic change within each tumor) correlated with improved responses to immune checkpoint inhibition. We also found that almost all cutaneous melanomas have a driver mutation in the MAPK pathway, underscoring the key role of MAPK-targeted therapies in melanoma drug development.

**Abstract:**

Background: Cutaneous melanoma (CM) can be molecularly classified into four groups: *BRAF* mutant, *NRAS* mutant, *NF1* mutant and triple wild-type (TWT) tumors lacking any of these three alterations. In the era of immune checkpoint inhibition (ICI) and targeted molecular therapy, the clinical significance of these groups remains unclear. Here, we integrate targeted DNA sequencing with comprehensive clinical follow-up in CM patients. Methods: This was a retrospective cohort study that assessed clinical and molecular features from patients with localized or metastatic CM who underwent targeted next-generation sequencing as part of routine clinical care. A total of 254 patients with CM who had a CLIA-certified targeted sequencing assay performed on their tumor tissue were included. Results: Of the 254 patients with cutaneous melanoma, 77 were *BRAF* mutant (30.3%), 77 were *NRAS* mutant (30.3%), 47 were *NF1* mutant (18.5%), 33 were TWT (13.0%) and the remaining 20 (7.9%) carried mutations in multiple driver genes (*BRAF*/*NRAS*/*NF1* co-mutated). The majority of this co-mutation group carried mutations in *NF1* (*n* = 19 or 90%) with co-occurring mutations in *BRAF* or *NRAS,* often with a weaker oncogenic variant. Consistently, *NF1* mutant tumors harbored numerous significantly co-altered genes compared to *BRAF* or *NRAS* mutant tumors. The majority of TWT tumors (*n* = 29, 87.9%) harbor a pathogenic mutation within a known Ras/MAPK signaling pathway component. Of the 154 cases with available TMB data, the median TMB was 20 (range 0.7–266 mutations/Mb). A total of 14 cases (9.1%) were classified as having a low TMB (≤5 mutations/Mb), 64 of 154 (41.6%) had an intermediate TMB (>5 and ≤20 mutations/Mb), 40 of 154 (26.0%) had a high TMB (>20 and ≤50 mutations/Mb) and 36 of 154 (23.4%) were classified as having a very high TMB (>50 mutations/Mb). *NRAS* mutant melanoma demonstrated significantly decreased overall survival on multivariable analysis (HR for death 2.95, 95% CI 1.13–7.69, *p* = 0.027, log-rank test) compared with other TCGA molecular subgroups. Of the 116 patients in our cohort with available treatment data, 36 received a combination of dual ICI with anti-CTLA4 and anti-PD1 inhibition as first-line therapy. Elevated TMB was associated with significantly longer progression-free survival following dual-agent ICI (HR 0.26, 95% CI 0.07–0.90, *p* = 0.033, log-rank test). Conclusions: *NRAS* mutation in CMs correlated with significantly worse overall survival. Elevated TMB was associated with increased progression-free survival for patients treated with a combination of dual ICI, supporting the potential utility of TMB as a predictive biomarker for ICI response in melanoma.

## 1. Introduction

Cutaneous melanoma (CM) is associated with Ras mutations that activate downstream mitogen-activated protein kinase (MAPK) signaling [1,2,3,4]. CM can accordingly be molecularly classified based on the MAPK pathway molecular driver, which includes *BRAF* mutations, *NRAS* mutations and *NF1* deficiency, accounting for ~80% of tumors [2,4]. *BRAF*, *NRAS* and *NF1* are generally thought to be mutually exclusive oncogenic drivers, although co-occurring driver mutations can be seen, particularly with *NF1* loss [4]. *BRAF* and *NRAS* mutant melanomas histopathologically correlate with low cumulative sun damage (low-CSD) melanomas, while *NF1* mutant melanomas are classified as high-CSD [5]. The remaining 20% of melanomas are deemed “triple wild-type” (TWT) and are clinically comprised less common melanoma subtypes such as acral, mucosal and uveal melanomas occurring in sun-shielded locations (classified as non-CSD melanomas) and often driven by alternate pathways of melanomagenesis.

While driver group is associated with distinct demographic and pathologic features [4,6,7,8], the relationship between molecular driver, clinical outcomes and ICI response remains unclear. Prior large-scale genomic studies, including the TCGA landscape study of 333 primary and metastatic melanomas, showed no significant correlations between genomic classification and clinical outcome [4]. However, this was published prior to widespread ICI use, only one year after the approval of anti-PD1 agents in metastatic melanoma [9,10,11,12].

Therapeutic advances in melanoma have resulted in the FDA approval of immune checkpoint inhibitors and several generations of BRAF-targeted therapies for patients with *BRAF* V600E mutant tumors. Accordingly, *BRAF* mutant melanomas exhibit improved overall survival compared with other cutaneous melanomas [13]. *BRAF* mutation was independently associated with improved recurrent-free survival (RFS) in patients with stage three melanoma who were treated with adjuvant pembrolizumab [14]. In addition, patients with *BRAF* mutations were shown to have superior outcomes when treated with first-line ICI compared with BRAF and MEK inhibition, and this has now become standard practice for a majority of patients [15]. Improved survival outcomes in *BRAF* mutant melanomas may be related to disease biology in addition to the availability of multiple therapeutic avenues.

Pre-clinical work and retrospective genomic studies have implicated *NRAS* mutation as an overall poor prognostic factor, with *NRAS* mutations associated with increased Breslow thickness [16,17]. However, data correlating clinical outcomes and responses to ICI for *NRAS* mutant melanoma are mixed [16,18,19,20]. The phase III IMspire170 trial compared cobimetinib + atezolizumab to pembrolizumab in patients with *BRAF* wild-type melanoma and found no differences in PFS or ORR based on *NRAS*, *NF1* or TWT molecular drivers in either of the two study arms [21]. In a phase II trial evaluating ipilimumab vs. ipilimumab/nivolumab, *NRAS* mutations were noted to be enriched in patients who experienced clinical benefit [18], suggesting that molecular driver disease biology may affect ICI response in this patient population. With regard to other biomarkers of ICI response, the only robustly associated tumor-specific molecular feature is tumor mutational burden (TMB) [22]. A high TMB correlates with improved ICI response and overall survival across a range of cancer types, including melanoma [23]. Given that almost all cutaneous melanomas have an intermediate or high TMB, it remains unclear whether there exists a TMB threshold above which the predictive significance of this biomarker becomes less robust.

Taken together, there remains a critical need to determine the precise relationship between molecular–genetic tumor alterations and clinical outcomes with contemporaneous systemic therapy regimens [24]. Here, we retrospectively identified a real-world cohort of 254 patients with CM undergoing targeted DNA sequencing of recurrently mutated cancer to evaluate whether any tumor-specific genomic features, particularly TCGA driver and tumor mutational burden, correlated with clinical outcomes in the real-world setting. We additionally assessed alternate mutations in TWT sun-exposed cutaneous melanomas to better understand this under-studied and poorly understood cutaneous melanoma phenotype. We found that *NRAS* mutation was associated with poorer overall survival (OS), and TMB was predictive of dual ICI response, suggesting that tumor genetics may help serve as an informative biomarker predictive of ICI response and clinical outcomes for cutaneous melanoma.

## 2. Methods

### 2.1. Cohort Selection/Identification and Targeted DNA Sequencing Analysis

A total of 330 patients with melanoma who underwent in-house-targeted next-generation DNA sequencing were retrospectively identified using the UCSF tumor registry, from which medical records, baseline demographics and pathologic and clinical outcome data were extracted. Uveal, acral, mucosal, primary CNS/meningeal and pediatric melanomas were excluded. Melanomas of unknown primary (*n* = 68) sites were assumed to be predominantly cutaneous and were included, leading to a total of 254 cases included for subsequent analysis. All cases were re-reviewed by a board-certified pathologist for inclusion in the present cohort. DNA sequencing was performed at the UCSF Clinical Cancer Genomics Laboratory (CCGL) using the UCSF500 (https://genomics.ucsf.edu/UCSF500, accessed on 1 May 2022) CLIA-certified and targeted DNA next-generation sequencing assay obtained as part of routine clinical care. Briefly, this assay uses a custom bait library (Roche Nimblegen) to cover the genomic sequence of 529 cancer-related genes and select introns of 47 genes. The high throughput sequencing of captured libraries was performed using Illumina NovaSeq6s000, aligned to the human reference genome, and variants were called using an internal pipeline and then filtered before undergoing manual review to assign functions based on known pathogenic alterations and predicted effects on proteins by a team of board-certified pathologists and the UCSF CCGL team. Microsatellite instability (MSI) was quantified from targeted sequencing data using MSIsensor [25]. A cutoff of 30% or greater was used to classify tumors as “MSI-high” per our UCSF500 clinical testing protocol. Given that the highest percentage of microsatellite instability in the cohort was 4.7%, none of the cases were deemed MSI-high.

### 2.2. Outcomes

The primary outcome assessed in this retrospective cohort was overall survival from the time of initial melanoma diagnosis. Secondary outcomes included progression-free survival evaluated in the context of systemic therapy for metastatic disease and recurrence-free survival for patients with localized disease treated with surgical intervention +/− adjuvant therapy. For the purposes of this study, progression-free survival was defined as the time from initiation of first-line systemic therapy to disease progression or death from any cause. Recurrence-free survival was defined as the time from the date of primary resection for localized disease to the time of recurrent melanoma or death from any cause. Patients who developed recurrent disease and then subsequently received systemic therapy were included in both the recurrence-free survival and progression-free survival analysis. Detailed information regarding the systemic therapy regimen and timing for metastatic disease is available for 116 patients (Table 1). The majority of these patients received immunotherapy as first-line systemic therapy (*n* = 108, 93.1%), with 8 patients receiving first-line BRAF/MEK inhibition. Outcomes were assessed separately for patients who received first-line anti-PD1 (*n* = 65) and for patients who received a first-line combination of dual ICI (anti-CTLA4 + anti-PD1, *n* = 36), as well as for all patients who received any form of immunotherapy in the first-line metastatic setting (including rare cases of anti-CLTA4 monotherapy, *n* = 7).

### 2.3. Statistical Analysis

An evaluation of categorical variables such as sex, ECOG status, stage, TMB group and TCGA driver gene was performed using either the chi-squared or Fisher’s exact test. Fisher’s exact test was used to assess proportions within two categorical variables. A chi-squared test was used to assess proportions within three or more categorical variables. An unpaired *t*-test was used forcomparison of continuous variables such as age, TMB and MSI (%). All hypothesis tests were 2-sided and considered significant at a value < 0.05. Univariable Cox Proportional Hazards (CPHs) were performed using STATA to evaluate overall survival, progression-free survival, recurrence-free survival and the following variables: age, sex, TMB, MSI (%), ECOG, presence of CNS disease, stage, TCGA mutation driver group (assessed independently for *BRAF*, *NRAS*, *NF1*, TWT and co-mutation group), any *NF1* mutation and TERT mutation status. Any variables found to have *p* < 0.1 in univariable CPH analysis were included in a multivariable CPH model for overall survival (OS), progression-free survival (PFS) and recurrence-free survival (RFS). Kaplan–Meier curves were then generated for OS, PFS and RFS using PRISM.

### 2.4. Co-Mutation Analysis

In an effort to uncover other genes that were commonly altered together in this patient cohort, a co-mutation analysis was also performed. Each gene was first examined for the frequency at which it was altered across all patients, and the resulting gene list was filtered to only include the top 5% of genes that were most commonly altered. The ensuing threshold of ≥40 alterations resulted in 30 candidate genes for co-mutation analysis. Patients were then examined for the presence of alterations in these 30 genes, and the associations between genes were subsequently examined via Pearson’s correlation analysis. *p*-values were adjusted for multiple comparisons using the Bonferroni correction method, and significance was defined as an adjusted *p*-value < 0.05. All co-mutation analyses were conducted in R (R Foundation for Statistical Computing), version 4.2.2.

## 3. Results

### 3.1. Molecular Melanoma Groups Display Characteristic Phenotypic and Demographic Features

The baseline demographic features for the 254 patients with cutaneous melanoma who underwent targeted DNA sequencing are summarized in Table 1. Targeted sequencing was performed at the time of initial melanoma diagnosis in 103/225 (45.8%) of cases and performed at a later time point in clinical care for melanoma in 122/225 cases (54.2%). The majority of cases (136/232, 58.6%) were metastatic at the time of tissue sampling for next-generation sequencing. The median age for the overall cohort was 61 (range 20–100). Patients were predominantly male (*n* = 170/253, 67.2%), consistent with previously reported demographics for cutaneous melanoma in the United States [26]. The median follow-up time for all patients was 39 months (IQR 13–79 months). A total of 186 cases were confirmed cutaneous melanoma (73.2%), while a total of 68 cases (26.8%) were melanoma of unknown primary (MUP) sites. MUPs were included in our analysis based on histopathologic review by board-certified dermatopathologists and prior data that suggest a majority of MUPs exhibit mutational profiles consistent with a sun-exposed cutaneous origin (Table 1) [27,28].

Cases were classified based on previously reported TCGA driver genes, including *BRAF* mutation (*n* = 77, 30.3%), *NRAS* mutation (*n* = 77, 30.3%), *NF1* mutation (*n* = 47, 18.5%) and TWT (*n* = 33, 13.0%) (Figure 1a) [4]. A small cohort of cases demonstrated mutations in multiple TCGA driver genes (*n* = 20, 7.9%), and within this subset, a majority harbored mutations in *NF1* as well as either *BRAF* or *NRAS* (*n* = 18 of 20, 90%) (Figure 1a). In addition to TCGA driver genes (*BRAF*, *NRAS*, *NF1*), we evaluated recurrent molecular alterations observed in at least 5% of the entire cohort, the majority of which were established oncogenes or tumor suppressors (Figure 1b) [29,30,31,32,33,34,35,36,37,38,39,40,41,42]. Recurrently altered genes included *TERT* (*n* = 216), *CDKN2A/B* (*n* = 116) *TP53* (*n* = 93), *ARID2* (*n* = 32), *PTEN* (*n* = 25), *RASA2* (*n* = 22), *PTEN* (*n* = 25), *NFKBIE* (*n* = 20), *KMT2B* (*n* = 16), *PTPRD* (*n* = 15), *PTPRT* (*n* = 15), *MAP2K1* (*n* = 14), *RAC1* (*n* = 14), *GRIN2A* (*n* = 13), *PTPRB (n* = 13 or 5%), *PREX2* (*n* = 13 or 5%), and *SETD2* (*n* = 13 or 5%). The three most commonly co-altered genes included activating *TERT* promoter mutations and loss of the tumor suppressors *CDKN2A/B* and *TP53*. A total of 121 cases demonstrated mutations in 2 of these 3 genes, and an additional 23 cases were “triple hit” in terms of alterations in *TERT*, *CDKN2A/B* and *TP53.* In addition, there were 25 cases harboring the *PTEN* alteration, of which 12 co-occurred with *CDKN2A/B* loss and 10 co-occurred with *TP53* mutation.

*BRAF* mutant melanoma tended to occur in younger patients with a median age of 54 (*p* = 0.0001, one-way ANOVA), which may account, in part, for previous reports of improved outcomes in *BRAF* mutant melanoma, while *NF1* mutant tumors were seen in older patients with a median age of 67 (*p* = 0.005, one-way ANOVA) compared with other driver genes (Figure 1d). Tumor mutational burden was significantly higher in *NF1* mutant tumors (median 70.2 mutations/Mb) and co-mutated tumors (median 66.5 mutations/Mb) (*p* < 0.001, one-way ANOVA) compared with other TCGA driver genes (Figure 1c). *BRAF* mutant tumors had the lowest median TMB (12.4 mutations/Mb), followed by *NRAS* mutant tumors (median 17.5 mutations/Mb) and TWT melanomas (28.2 mutations/Mb) (Figure 1c). Patients with *BRAF* mutant tumors were associated with increased rates of non-metastatic disease at diagnosis (*p* < 0.0001, chi-square test), while patients with TWT tumors were associated with poorer performance status (ECOG 1+) (*n* = 7/20, 35%) compared with other TCGA driver genes (*n* = 21/144, 14.6%; *p* = 0.028, Fisher’s exact test) (Appendix A). With regard to anatomic location, *NF1* mutant tumors (*n* = 22/47, 46.8%), TWT melanomas (*n* = 14/33, 42.4%) and co-mutant tumors (*n* = 7/20, 35%) occurred more frequently on the head/neck compared with *BRAF* or *NRAS* mutant tumors (*n* = 24/154, 15.6%; *p* < 0.001, chi-squared test) (Appendix A). There were otherwise no statistically significant differences in baseline features between TCGA driver genes (Table 1; Appendix A).

### 3.2. BRAF, NRAS and NF1 Mutant Melanomas Exhibit Distinct Alteration and Co-Mutation Patterns

*BRAF* mutations can be classified into the following three groups: Class I, II or III [43]. Class I and Class II mutations lead to Ras-independent Raf activation, with Class I mutations involving V600 to create monomers with high kinase activity and Class II mutations creating activated Raf dimers with intermediate kinase activity. In contrast, class III mutations comprise weaker oncogenic variants leading to enhanced Ras binding with impaired kinase activity that drives CRAF activation for enhanced downstream MAPK pathway signaling [44]. In our cohort, *BRAF* mutations were predominantly class I (Figure 2a), particularly for the *BRAF* driver group, in which 68 of 77 cases (88.3%) had class I *BRAF* mutations. Only one class III *BRAF* mutation was noted in this group (*BRAF* G466E), with several class II mutations [43]. Multiple fusions and gene rearrangements were also detected involving *AGK* and *BRAF* (*n* = 3) and *ATF7* and *BRAF* (*n* = 1). In contrast, only one of the 11 *BRAF* mutations identified in the *BRAF/NRAS/NF1* co-mutation group was a class I *BRAF* mutation. When occurring in the co-mutation group, the majority (8 of 11 or 73%) of *BRAF* mutations were class III, consistent with prior work suggesting these weaker oncogenic *BRAF* variants require additional hits to drive melanomagenesis [43].

*NRAS* mutations were predominantly Q61 variants regardless of whether the tumors were in the *NRAS* driver (*n* = 67/77 or 86%) or co-mutated group (*n* = 9/14 or 64.3%) (Figure 2b). A total of 67 putative loss of function variants in NF1 were identified, with the majority clustered at the 5′ end of the gene, suggesting that the upstream loss of function mutations is enriched in *NF1* (Figure 2c). There were no differences in specific *NF1* variants when co-mutated with other driver genes compared to tumors with the *NF1* mutation alone.

Correlation analysis of all identified genes found *NF1* mutations to correlate with several genomic alterations, including, most frequently, *RASA2* loss (correlation coefficient 0.37, adjusted *p* < 0.001) but also frequently with *PTPRT* mutations (correlation coefficient 0.33, adjusted *p* < 0.001), *TP53* mutations (correlation coefficient 0.31, adjusted *p* < 0.001), *PTPRB* mutations (correlation coefficient 0.30, adjusted *p* < 0.001) and 39 other less commonly co-mutated genes (Figure 2d,e and Appendix A). *NRAS* correlated most with *TERT* (correlation coefficient 0.33, *p* < 0.001), *BRAF* was not significantly co-altered with any genes examined, and *BRAF* and *NRAS* demonstrated significant mutual exclusivity (Figure 2d and Appendix A).

### 3.3. Triple Wild-Type Tumors Contain Frequent Alterations in Other Components of MAPK Pathway Signaling

The majority of TWT tumors demonstrated a pathogenic variant in the Ras pathway (*n* = 28/33 or 84.8%) (Figure 3a,b). These included *MAP2K1* (*n* = 7), *SPRED1* (*n* = 4), *KIT* (*n* = 3), *FGF4/19* (*n* = 3), *MAP3K8* (*n* = 2), *RASA2* (*n* = 2), *ERBB2* (*n* = 2), *MET* (*n* = 2), *NF2* (*n* = 2) and *GAB2* (*n* = 2). (Figure 3b,d). In tumors harboring a *BRAF, NRAS,* or *NF1* mutation (non-TWT tumors), *MAP2K1* mutations were identified in 7 of 221 (or 3.2%) tumors, *SPRED1* mutations were found in 5 of 221 (or 2.3%) tumors, *KIT* in 7 of 221 (or 3.2%) tumors and *FGF4/19* in 10 of 221 (or 4.5%) tumors. In addition, triple wild-type tumors often harbored non-Ras pathway alterations found in other TCGA molecular groups including *TERT* (*n* = 21), *TP53* (*n* = 17), *CDKN2A/B* (*n* = 12), *NFKBIE* (*n* = 3), *YAP1* (*n* = 3), *FAT1* (*n* = 3), *NOTCH1* (*n* = 3), *NOTCH2* (*n* = 3), *PTEN* (*n* = 3), *MITF* (*n* = 3) and *ATM* (*n* = 3) (Figure 3c,d). A total of 5 of 33 TWT cases demonstrated no known or suspected Ras pathway alterations and demonstrated presumed driver mutations in *EIF1AX*, *GNAQ*, *ATRX* and *ASXL1* (Figure 3a). The *GNAQ* mutation was found in a melanoma of unknown primary site, while the *EIF1AX* mutation was found in a cutaneous melanoma located in the neck.

### 3.4. NRAS Mutation Is Prognostic, and Tumor Mutational Burden (TMB) Is Predictive of Response to Dual Checkpoint Blockade

We next assessed clinical outcomes for the entire 254-patient cohort. With regard to overall survival (OS), univariable CPH analysis revealed that age, ECOG status, CNS disease, stage, *BRAF* mutation and *NRAS* mutation were significantly correlated with OS (*p* < 0.1; Appendix A). For multivariable analysis, *NRAS* mutant tumors remained correlated with worse OS (HR 2.95, 95% CI 1.13–7.69, *p* = 0.027; log-rank test) (Figure 4a, Appendix A). Older age (HR 1.06, 95% CI 1.02–1.11, *p* = 0.003; log-rank test) and higher ECOG scores (HR 4.84, 95% CI 2.12–11.05, *p* < 0.001; log-rank test) also correlated with decreased overall survival (Figure 4b, Appendix A). Tumor mutational burden did not correlate with overall survival in our cohort (Appendix A), and the TCGA molecular group was not associated with differences in progression-free survival or recurrence-free survival (Appendix A). Multivariable analysis for recurrence-free survival found age to be the only variable correlated with RFS (HR 1.03, 95% CI 1.02–1.05, *p* < 0.001, log-rank test) (Appendix A).

Of the 116 patients treated with systemic therapy for metastatic or locally advanced/unresectable disease, the majority were treated with first-line ICI: 65 out of 116 (56.0%) patients received anti-PD1 monotherapy, 36 of 116 (31.0%) patients received dual checkpoint inhibition with a combination of anti-PD1 and anti-CTLA4, and 7 of 116 patients (6.0%) received anti-CTLA4 monotherapy (Table 1). A small proportion were treated with BRAF/MEK inhibitors in the first-line setting (*n* = 8/116 or 6.9%). Multivariable analysis found no significant associations with progression-free survival on any first-line immune checkpoint inhibition (Appendix A). There was a trend towards improved progression-free survival for patients with mutations in at least 2 of the 3 most commonly mutated tumor suppressor genes (TERT, CKDN2A/B and TP53) with HR for progressions of 0.613 (95% CI 0.373–1.009, *p* = 0.054), although this did not reach statistical significance. When further stratified based on specific treatment type, TMB was the only variable significantly associated with response to dual ICI; no variables were associated with response to single-agent anti-PD1 (Figure 4c, Appendix A). A low TMB (≤5 mutations/Mb) correlated with poorer PFS in patients receiving dual-agent ICI (HR 20.9, 95% CI 1.77–245.9; *p* = 0.016; log-rank test) (Figure 4d, Appendix A). Given the overall small sample size, particularly in the low TMB cohort, which included only 14 patients, this relationship between TMB and clinical outcome requires further corroboration with larger, ideally multi-institutional datasets.

## 4. Discussion

We present clinical and molecular data from 254 patients with melanoma who underwent targeted DNA sequencing and were treated in the ICI era. Our analysis confirmed CM classification based on TCGA genetic drivers [4] and previously reported associated demographic and phenotypic features, including age of onset (*BRAF* mutations in younger patients and *NF1* mutations found in older patients) and significantly higher median TMB in *NF1* mutant tumors [4,8] (Figure 1c,d). *BRAF* mutations in the entire 254-patient cohort were predominantly class I (*n* = 68 of 88 or 77.3%) (Figure 2a). While cases with *BRAF* TCGA driver mutations alone were consistent with previously reported data suggesting a predominance of class I mutations in *BRAF* mutant melanoma, only one of eleven *BRAF* co-mutated cases (co-occurring with either *NRAS* or *NF1*) exhibited a class I *BRAF* mutation. The majority of co-mutated cases had weaker class III *BRAF* mutations, which have been associated with improved prognosis in melanoma compared with class I *BRAF* mutations [45]. We found *NF1* to be the only TCGA driver in our cohort that significantly correlated with multiple other genes, most frequently *RASA2*, as previously reported [46]. *NF1* co-mutations with either *BRAF* or *NRAS*, as well as frequently with other melanoma-associated oncogenes, suggest that this is overall a weaker oncogenic driver that may cooperate with additional MAPK pathway alterations to drive melanomagenesis. *NRAS* mutant tumors correlated with poorer overall survival compared with other TCGA driver gene groups. The presence of an *NRAS* mutation has historically been correlated with poorer overall survival in patients with advanced melanoma, but its prognostic impact in the modern treatment era is less clear [47,48]. Our analysis of patients treated in the immunotherapy era (from 2013–2023) found *NRAS* mutation to be an independent factor significantly associated with poorer overall survival, underscoring the need for further drug development in this particular patient population.

The majority of TWT tumors carried at least one putative driver mutation involving the MAPK pathway, including *MAP2K1*, *SPRED1*, *KIT*, *FGF4/19*, *RASA2* and *ERBB2* (Figure 3). Triple wild-type tumors also demonstrated a median TMB of 28.2, suggesting that the correlation between low TMB and triple wild-type status may be most relevant to sun-shielded or non-cutaneous melanomas (Figure 1c). Only 5 of 33 TWT melanomas in our cohort lacked a putative driver mutation involving the MAPK pathway. Our analysis suggests that the majority of TWT cutaneous melanomas harbor a Ras/MAPK pathway mutation, with some confirmed cutaneous melanomas demonstrating mutations more commonly identified in uveal melanoma, including *EIF1AX* and *GNAQ*.

Beyond TCGA drivers, several studies have focused on identifying additional prognostic or predictive tumor-specific genomic aberrations in cutaneous melanoma, particularly in terms of ICI response. Mutations in *KMT2*, *PTPRT*, *MAP2K1* and *SETD2* have been associated with improved responses to immune checkpoint inhibitors, while *PTEN* loss of function is thought in some studies to be associated with innate resistance to immune checkpoint blockade [49,50,51,52,53]. We did not find any specific genomic alterations other than the presence of an *NRAS* mutation to correlate with clinical outcomes in our 254-patient cohort. TMB was the only predictive biomarker of response to immune checkpoint inhibition identified in our analysis. Notably, TMB was not associated with PFS on single-agent anti-PD1 but was predictive for patients treated with dual-agent immune checkpoint inhibition in our patient population. Interestingly, despite a significantly higher TMB than other driver groups, *NF1* mutant and co-mutant tumors did not demonstrate improved PFS when treated with first-line immunotherapy. This may be related to sample size and the wide range of TMB findings within the *NF1* mutant group. Given that a higher TMB has been correlated with response to ICI, both in our study and in several others, and TMB in *NF1* mutant melanomas is generally very high, further investigation is warranted as to why *NF1* mutation alone did not correlate with improved ICI response.

## 5. Conclusions

Our real-world retrospective analysis of 254 patients with melanoma who underwent in-house DNA sequencing at our institution generated several novel and notable findings. *NRAS* mutant tumors were associated with worse overall survival outcomes. We also demonstrate that triple wild-type cutaneous melanomas that lack mutations in *BRAF*, *NRAS* or *NF1* have a similar TMB to the overall median and, in the majority of cases, demonstrate putative drivers in the MAPK pathway, suggesting that disease biology in these cases is likely similar to those melanomas with known Ras drivers. Finally, TCGA driver group was not associated with progression-free survival on any first-line immunotherapy despite prognostic significance in terms of overall survival. Tumor mutational burden was associated with response to a combination of immune checkpoint inhibition with anti-CTLA4 and anti-PD1 but did not have a statistically significant association with progression-free survival on anti-PD1 monotherapy. Taken together, these data support prognostic differences based on driver mutation and the predictive utility for TMB in cutaneous melanoma. Additional investigation into “triple wild-type” cutaneous melanomas and their reliance on Ras/MAPK signaling is an area for further study, and poor outcomes in *NRAS* mutant tumors highlight an urgent, unmet clinical need for therapeutic development.

Our study has several limitations, most notably that it is a retrospective single-institution investigation with all the caveats such an approach entails. In addition, detailed follow-up and treatment data were available only for a subset of the 254 patients that underwent molecular profiling, and both metastatic and non-metastatic sites were included with different patterns of follow-up, making results difficult to meaningfully interpret and apply to distinct clinical populations. Further prospective studies that assess the prognostic significance of driver mutations and the predictive and prognostic significance of tumor mutational burden are certainly warranted.

## Figures and Tables

**Figure 1 cancers-16-01347-f001:**
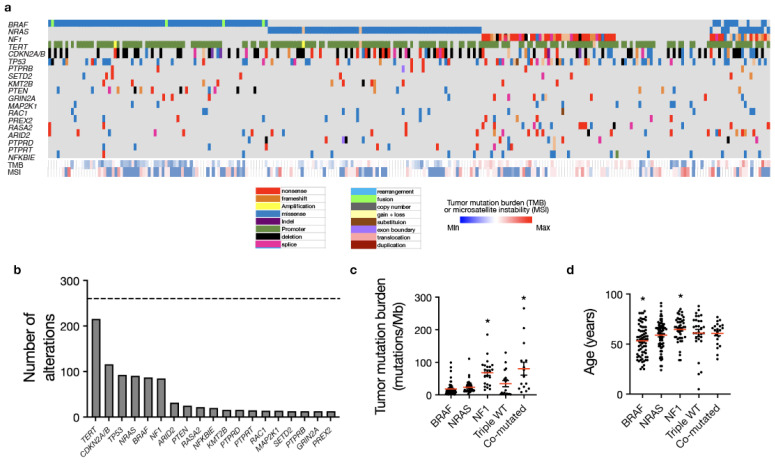
Mutational analysis of metastatic melanoma from a real-world cohort recapitulates TCGA molecular drivers. (**a**) Targeted DNA sequencing analysis using a CLIA-certified clinical genomics assay of 254 patients undergoing clinical care at a tertiary cancer center reveals the following five molecular groups: *BRAF* mutant (*n* = 77), *NRAS* mutant (*n* = 77), *NF1* mutant (*n* = 47), triple wild-type (*n* = 33), and co-mutated tumors with multiple alterations in *BRAF*, *NRAS*, and/or *NF1* (*n* = 20) consistent with TCGA molecular groups. (**b**) A total of 19 genes were recurrently altered in at least 5% of the cutaneous melanoma cohort including *TERT* (*n* = 216), *CDKN2A/B* (*n* = 116), *TP53* (*n* = 93), *NRAS* (*n* = 91), *BRAF* (*n* = 87), *NF1* (*n* = 85), *ARID2* (*n* = 32), *PTEN* (*n* = 25), *RASA2* (*n* = 22), *NFKBIE* (*n* = 20), *KMT2B* (*n* =16), *PTPRD* (*n* = 15), *PTPRT* (*n* = 15), *RAC1* (*n* = 14), *MAP2K1* (*n* = 14), *SETD2* (*n* = 13), *PTPRB* (*n* = 13), *GRIN2A* (*n* = 13) and *PREX2* (*n* = 13). (**c**) Median tumor mutation burden (shown by the red lines) is significantly increased in *NF1* mutant and co-mutated tumors (as denoted by the *) compared to *BRAF*, *NRAS*, or TWT tumors (ANOVA, *p* < 0.001). (**d**) Patients with *BRAF* mutant tumors are significantly younger, while *NF1* mutant patients are significantly older than other molecular groups (ANOVA, *p* = 0.001).

**Figure 2 cancers-16-01347-f002:**
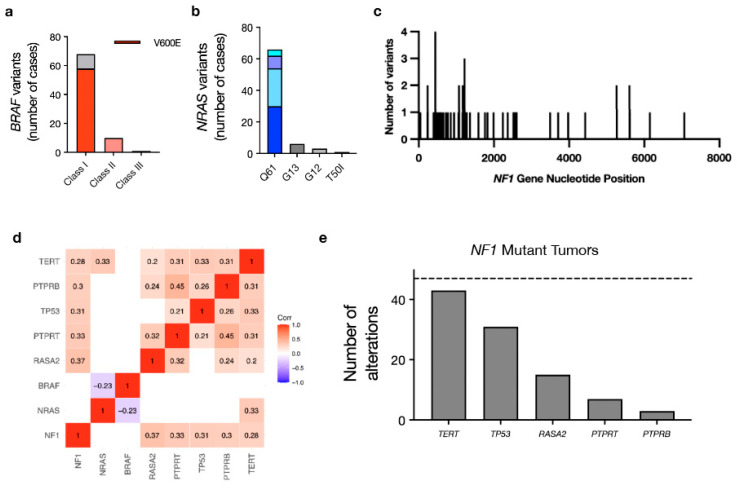
Driver mutation-specific analysis of BRAF, NRAS and NF1 mutant tumors reveals specific variant and co-mutation patterns. (**a**) *BRAF* mutant tumors were primarily composed of class I *BRAF* mutations (*n* = 68 or 86%). The majority were V600E (*n* = 58, shown in red) with 8 V600K mutations and 2 V600R mutations. A total of 3 *BRAF* fusions were detected, as well as 2 gene re-arrangements, most of which involved *AGK* and *BRAF*. Other non-class I loci in this group included K601, G469, G466, F247 and E586. There were no associated differences in outcomes with class I mutations compared to other *BRAF* classes or to non-*BRAF* mutant tumors. Only 1 of 10 *BRAF* mutant cases with co-occurring *NRAS* or *NF1* mutations (10%) had a class I *BRAF* mutation. S467L mutations were particularly common in this cohort (*n* = 4 or 40%). Other loci included N581, G469, L597 and N580. (**b**) The majority of *NRAS* mutant tumors had mutations at Q61 (N = 66 or 87%) including Q61R (*n* = 30), Q61K (*n* = 24), Q61L (*n* = 8) and Q61H (*n* = 4). These correspond to color coded bars in the figure. Other mutations identified included G13R (*n* = 5), G12A (*n* = 3), G13C (*n* = 1) and T50I (*n* = 1). Two amplifications were noted, as well as one case with copy-number-neutral LOH at Q61K. Of the 15 *NRAS* mutations identified in the co-mutation group, 10 were Q61K/L/R (67%), 2 were G12A/D, 2 were G13D, and 1 was T50I. (**c**) A wide spectrum of variants, enriched on the 5′ ends of the gene, was identified in *NF1* mutant tumors, with 12 of 47 cases in the *NF1* driver group (26%) and 8 of 18 *NF1* mutant cases (44%) in the *BRAF/NRAS/NF1* group having more than one mutation in *NF1*. The presence of multiple *NF1* mutations did not have a significant impact on overall survival when compared with tumors with only one *NF1* alteration. The majority of the *BRAF/NRAS/NF1* co-mutant tumor group had mutations in *NF1* (*n* = 18 or 90%). There were no significant survival outcome differences between this group and the *NF1* driver mutation group. (**d**) Co-mutation correlation analysis (threshold Pearson > 0.3 at *p* < 0.05) found that *BRAF* mutations did not significantly correlate with any other alterations, *NRAS* mutations correlated with *TERT* mutations and *NF1* mutations were significantly correlated with several genes, including *RASA2*, *PTPRT*, *TP53*, *PTPRB* and *TERT*. Notably, *BRAF* and *NRAS* mutations were anti-correlated, consistent with their mutual exclusivity as drivers. (**e**) The co-mutation breakdown of *NF1* mutant tumors and significantly co-occurring alterations.

**Figure 3 cancers-16-01347-f003:**
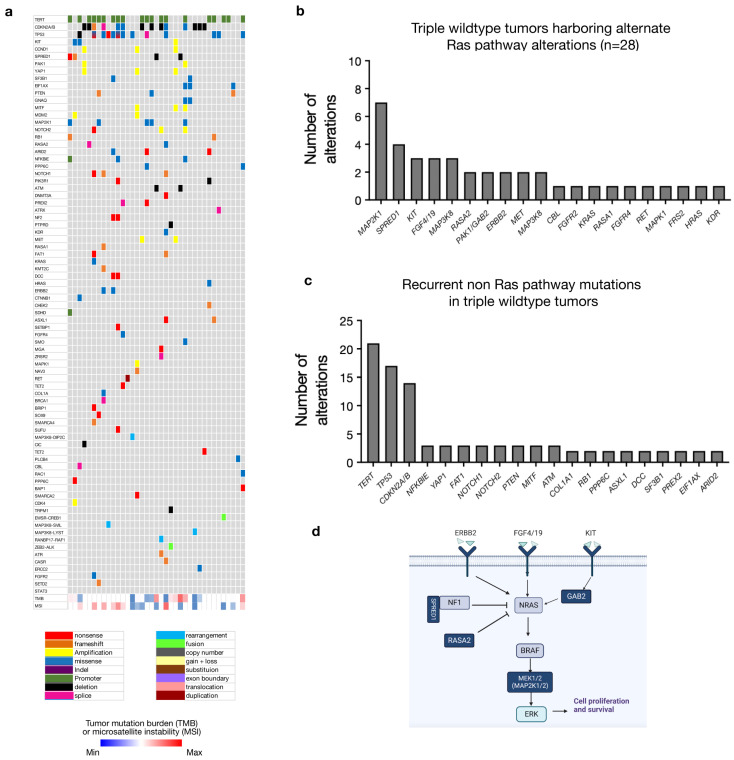
Triple wild-type tumors harbor alternate Ras pathway alterations in the majority of tumors lacking a classical TCGA driver mutation. (**a**) In triple wild-type tumors, recurrently altered genes include *TERT* (*n* = 21), *TP53* (*n* = 18), *CDKN2A/B* (*n* = 14), *MAP2K1* (*n* = 7) and *SPRED1* (*n* = 4). (**b**) In 28 of 33 cases (84.8%), pathogenic variants in known Ras pathway components are observed as putative driver mutations. (**c**) Recurrent alterations in non-Ras pathway genes in the triple wild-type group are predominantly the *TERT* and *TP53* mutation, similar to other molecular groups, with additional mutations comprising potential drivers in tumors without a classic Ras pathway mutation. (**d**) Schematic depicting the mutations occurring in triple wild-type tumors (dark blue) within the Ras signaling pathway.

**Figure 4 cancers-16-01347-f004:**
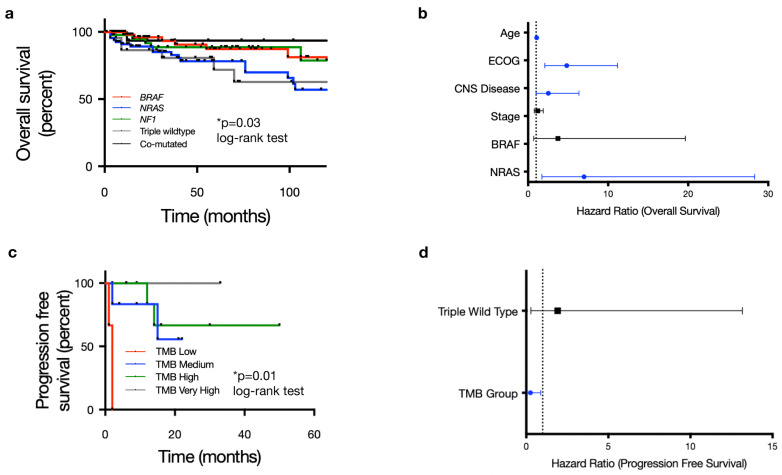
Molecular subgroup is prognostic for overall survival, and tumor mutation burden (TMB) is predictive of dual checkpoint blockade response in cutaneous melanoma. (**a**) Overall survival stratified by 5 pre-specified molecular subgroups (*BRAF* mutant, *NRAS* mutant, *NF1* mutant, triple wild-type and co-mutated *BRAF*/*NRAS*/*NF1*) shows statistically significant differences in outcome based on molecular driver (*p* = 0.03) with *NRAS* mutation associated with particularly worse outcomes. (**b**) Forest plot of multivariable hazard ratios for OS including covariates with *p* ≤ 0.1 in univariable survival analysis (ECOG status, presence of CNS disease, stage, BRAF, NRAF). In addition to *NRAS* mutation, age, ECOG status, and the presence of CNS disease were independent factors significantly associated with overall survival. (**c**) Kaplan–Meier curve depicting progression-free survival stratified by tumor mutation burden (TMB) for 36 out of 116 patients (31.0%) who received the combination of anti-PD1/PDL1 + anti-CTLA4 as the first-line systemic therapy for metastatic disease. This showed a statistically significant difference in PFS based on TMB group (*p* = 0.01). (**d**) TMB was the only covariate found to significantly impact PFS on combination ICI in multivariable CPH analysis.

**Table 1 cancers-16-01347-t001:** Baseline clinical characteristics of patients with metastatic melanoma undergoing target DNA sequencing (*n* = 254 total patients).

Parameter	Value
**Median Age** (range)	61 years (20–100 years)
Sex	
Female	83/254 (32.8%)
Male	170/254 (67.2%)
ECOG	
0	136/164 (82.9%)
1	24/164 (14.6%)
2	4/164 (2.4%)
CNS disease (ever developed)	
Yes	89/227 (39.2%)
No	138/227 (60.8%)
Primary Location	
Unknown	68/254 (26.8%)
Scalp/face/neck	67/254 (26.4%)
Back	39/254 (15.4%)
Lower extremity	36/254 (14.2%)
Upper extremity	26/254 (10.2%)
Chest	16/254 (6.3%)
Other	2/254 (0.8%)
NGS at Time of Initial Diagnosis?	
Yes	103/225 (45.8%)
No (performed at later time point)	122/225 (54.2%)
Metastatic (at time of NGS tissue sampling?)	
Yes	136/232 (58.6%)
No	96/232 (41.4%)
Median follow-up (IQR)	39 months (13–79 months)
TCGA Driver Mutation Groups	
BRAF mutant	77/254 (30.3%)
NRAS mutant	77/254 (30.3%)
NF1 mutant	47/254 (18.5%)
Triple wild-type	33/254 (13.0%)
BRAF/NRAS/NF1 co-mutated	20/254 (7.9%)
Median TMB (range)	20.0 (0.7–266 mutations/Mb)
TMB Group	
Low TMB (≤5 mutations/Mb)	14/154 (9.1%)
Intermediate TMB (>5 and ≤20 mutations/Mb)	64/154 (41.6%)
High TMB (>20 and ≤50 mutations/Mb)	40/154 (26.0%)
Very high TMB (>50 mutations/Mb)	36/154 (23.4%)
First-Line Systemic Therapy	
Anti-PD1/PDL1	65/116 (56.0%)
Anti-PD1/PDL1 + anti-CTLA4	36/116 (31.0%)
Anti-CTLA4	7/116 (6.0%)
BRAF/MEK inhibitors	8/116 (6.9%)
Adjuvant Therapy (*n* = 37 cases)	
Anti-PD1/PDL1	21/37 (56.8%)
Radiation	8/37 (21.6%)
Interferon	3/37 (8.1%)
Anti-CTLA4	2/37 (5.4%)
BRAF/MEK inhibitors	1/37 (2.7%)
Other	1/37 (2.7%)

## Data Availability

All data is available in Appendix A, and more detailed information, specifically clinical annotations, can be obtained by contacting the corresponding author.

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
