# Peer review of "Targeted DNA Sequencing of Cutaneous Melanoma Identifies Prognostic and Predictive Alterations"

_cancers, 2024, doi:10.3390/cancers16071347_

Round 1

Reviewer 1 Report

Comments and Suggestions for Authors

Dear authors, congratulations for your paper. An interesting subject that aims to shed some light on a very important topic: the phenotype of melanoma and it's response to modern treatment. I found your work very interesting, well designed, good statistics, great discussions and very interesting conclusion. I am for publication in it's present form.

Reviewer 2 Report

Comments and Suggestions for Authors

 The authors evaluated in a retrospective cohort of 254 cutaneous melanoma patients, with NGS analysis for targeted mutations.  The primary outcome was OS,  PFS/RFS were secondary outcomes according to the mutation group (BRAF, NRAS, NF1, TWT), to the TMB group (low, intermediate, high, very high), and, to the clinical characteristics as age, ECOG, CNS involvement and treatment.

Their conclusions are

1)  NRAS mutant melanoma had a significantly decreased survival, on multivariate analysis, compared to other TCGA molecular subgroups, however, ECOG score remains the more powerful predictor of OS.

2) Elevated TMB was associated with an increased PFS, however only for the 36 patients treated with IO.

3) Low TMB was associated with the poorest PFS (HR 20.9) in patients treated with IO.

Only 14 patients had a low TMB and 36 have been treated with IO in the whole series, indeed the CI =1.77-245.9 appears not convincing.

In the same way, the statement from lines 390-395 regarding the high TMB in NF1 melanoma and the absence of improved PFS with immunotherapy could be related to the low number of patients (47) and the wide dispersion of TMB value in this cohort (fig 1C).

The decision to analyze metastatic and no-metastatic patients together, the different adjuvant treatments applied, calculation of RFS from diagnosis to recurrence with different intervals, till six months, between diagnosis and adjuvant therapy, make less reliable and, transferable the results.

The statement on lines 77-78 is referred to a series of patients quite different, treated with nivolumab (94%) and, pembrolizumab (6%), moreover,  fig 2, reference 14, indicates that melanoma BRAF mutated with high TMB are only 5 and I would be cautious  in deriving conclusions from five patients

The S3a figure actually,   confirms the not significant impact of TMB on OS, as Mutational groups have any effect on PFS (FigS3b) as well on RFS (Fig S3c).

I agree with the authors that we need additional investigations on TWT, and NRAS cutaneous melanomas, with the hope of developing more active treatments and  improving the actual dismal prognosis-, for this purpose I appreciate their effort  in this field.

Author Response

Reviewer 2:

The authors evaluated in a retrospective cohort of 254 cutaneous melanoma patients, with NGS analysis for targeted mutations.  The primary outcome was OS, PFS/RFS were secondary outcomes according to the mutation group (BRAF, NRAS, NF1, TWT), to the TMB group (low, intermediate, high, very high), and, to the clinical characteristics as age, ECOG, CNS involvement and treatment.

Their conclusions are

1)  NRAS mutant melanoma had a significantly decreased survival, on multivariate analysis, compared to other TCGA molecular subgroups, however, ECOG score remains the more powerful predictor of OS.

2) Elevated TMB was associated with an increased PFS, however only for the 36 patients treated with IO.

3) Low TMB was associated with the poorest PFS (HR 20.9) in patients treated with IO.

Only 14 patients had a low TMB and 36 have been treated with IO in the whole series, indeed the CI =1.77-245.9 appears not convincing.

In the same way, the statement from lines 390-395 regarding the high TMB in NF1 melanoma and the absence of improved PFS with immunotherapy could be related to the low number of patients (47) and the wide dispersion of TMB value in this cohort (fig 1C).

This point is well taken, and we agree the limited sample size within the low TMB group and the lack of available treatment data for a significant proportion of our overall cohort is a limitation of the present work.  Accordingly, we have added additional language regarding the limited sample size on page 16 of the manuscript as follows: “Given the overall small sample size, particularly in the low TMB cohort which included only 14 patients, this relationship between TMB and clinical outcome requires further corroboration with larger, ideally multi-institutional datasets. data sets.”

In addition, we have added a line in the discussion regarding the TMB dispersion within NF1 mutant tumors, which may indeed be due in part to our sample size, on page 17 as follows: “This may be related to sample size and the wide range of TMB findings within the NF1 mutant group.”

The decision to analyze metastatic and no-metastatic patients together, the different adjuvant treatments applied, calculation of RFS from diagnosis to recurrence with different intervals, till six months, between diagnosis and adjuvant therapy, make less reliable and, transferable the results.

We agree and acknowledge this is a major limitation of our single institution retrospective series. In which patients were managed at the discretion of the discretion of the treating provider. We have amended our language in the conclusion acknowledging this on page 18 as follows “In addition, detailed follow-up and treatment data were available only for a subset of the 254 patients that underwent molecular profiling and both metastatic and non-metastatic sites were included with different patterns of follow-up, making results difficult to meaningfully interpret and apply to distinct clinical populations.”

The statement on lines 77-78 is referred to a series of patients quite different, treated with nivolumab (94%) and, pembrolizumab (6%), moreover fig 2, reference 14, indicates that melanoma BRAF mutated with high TMB are only 5 and I would be cautious in deriving conclusions from five patients.

This is a very reasonable point as well and important to acknowledge the limitations of a small sample size with inconsistent follow up, and we have added additional language in the conclusion acknowledging this sample size limitation in the results on page 15: “Given the overall small sample size, particularly in the low TMB cohort which included only 14 patients, this relationship between TMB and clinical outcome requires further corroboration with larger, ideally multi-institutional datasets. data sets. “ and in the conclusion on page 18 “In addition, detailed follow-up and treatment data were available only for a subset of the 254 patients that underwent molecular profiling and both metastatic and non-metastatic sites were included with different patterns of follow-up, making results difficult to meaningfully interpret and apply to distinct clinical populations.

The S3a figure actually confirms the not significant impact of TMB on OS, as Mutational groups have any effect on PFS (FigS3b) as well on RFS (Fig S3c).

This point is well taken, and we agree this should have been better described, as we neglected to highlight these findings in our initial submission.  We have now amended the results on page 15 as follows to explicitly highlight this point: “Tumor mutational burden did not correlate with overall survival in our cohort (Figure S3a), and TCGA molecular group was not associated with differences in progression free survival or recurrence free survival (Figure S3b-c).” 

I agree with the authors that we need additional investigations on TWT, and NRAS cutaneous melanomas, with the hope of developing more active treatments and improving the actual dismal prognosis-, for this purpose I appreciate their effort in this field.

Thank you for this positive feedback, and we very much are in agreement that both TWT and NRAS mutant melanomas are an unmet clinical need.

Reviewer 3 Report

Comments and Suggestions for Authors

The manuscript titled “Targeted DNA Sequencing of Cutaneous Melanoma Identifies Prognostic and Predictive Alterations” reports on mutations frequencies in cutaneous melanoma presenting an important but not overwhelming single institution retrospective cohort with limited treatment information. Though principally interesting, many points remain to be addressed before publication can be recommended.

Specific remarks:

In the simple summary, the authors write “In this study, we sought to understand how different genetic “drivers” in a group of 254 patients with sun-exposed cutaneous melanomas influenced overall survival and response to immunotherapy”. This evokes the impression that melanomas to be studied were selected for being sun-exposed but this is not clear from the rest of the manuscript. This should be clarified.

Uveal melanoma and melanocytic tumors of the CNS show a very distinct mutational profile and melanomas of unknown origin could easily be identified as belonging to one of the two classes and if they do, they should be excluded from further analysis. Are the TWT cases with mutations in GNAQ or EIF1AX of unknown origin?

The sequencing protocol and methods are not outlined.

How was MSI status established?

Penetrance (ratio of wt vs. mutated reads) of driver mutations and co-mutations should be analyzed.

Tumorigenic driver genes can also be hit by bystander mutations. The pathogenic potential of mutations in driver genes should be analyzed in the co-mutation group.

The mutual exclusivity of mutations of the three main tumor suppressor genes and features of tumors with double TSG hits should be analyzed.  TSG mutations and therapy response should be analyzed.

The frequency of MAP-kinase pathway mutations found in TWT cases should be analyzed for BRAF, NRAS and NF1 cases.

Author Response

Reviewer 3

The manuscript titled “Targeted DNA Sequencing of Cutaneous Melanoma Identifies Prognostic and Predictive Alterations” reports on mutations frequencies in cutaneous melanoma presenting an important but not overwhelming single institution retrospective cohort with limited treatment information. Though principally interesting, many points remain to be addressed before publication can be recommended.

Specific remarks:

In the simple summary, the authors write “In this study, we sought to understand how different genetic “drivers” in a group of 254 patients with sun-exposed cutaneous melanomas influenced overall survival and response to immunotherapy”. This evokes the impression that melanomas to be studied were selected for being sun-exposed but this is not clear from the rest of the manuscript. This should be clarified.

This is an excellent point, and one we should have been more careful to clarify. These melanomas were not selected specifically for being sun-exposed. We have thus omitted modifier in our amended summary, and the sentence now reads as follows:  “In this study, we sought to understand how different genetic “drivers” in a group of 254 patients with  cutaneous melanomas influenced overall survival and response to immunotherapy.”

Uveal melanoma and melanocytic tumors of the CNS show a very distinct mutational profile and melanomas of unknown origin could easily be identified as belonging to one of the two classes and if they do, they should be excluded from further analysis. Are the TWT cases with mutations in GNAQ or EIF1AX of unknown origin?

Thank you for the opportunity to better describe these cases in more detail, as the manuscript would benefit from further exposition. Upon detailed review with our pathology team, the GNAQ mutated case was classified as a melanoma of unknown primary while the EIF1AX mutant case was cutaneous melanoma in origin, located anatomically in the neck at time of initial diagnosis. We have added this information to the text on page 14: “The GNAQ mutation was found in a melanoma of unknown primary while the EIF1AX mutation was found in a  cutaneous melanoma located in the neck.”

While we acknowledge that melanomas of unknown primary may not all be of cutaneous origin, we assumed that all unknown primary were cutaneous and included these cases in our analysis for the present manuscript.  We did systemically exclude any primary CNS or uveal melanomasfrom this cohort.  However, after discussion with our multidisciplinary team of co-authors including two board certified pathologists specializing in dermatopathology, we felt it was most appropriate not to exclude unknown primaries based on molecular profile alone. We have included this information in the methods section on page 8 as follows: “Uveal, acral, mucosal, primary CNS/meningeal and pediatric melanomas were excluded.  Melanomas of unknown primary (n =68) were assumed to be predominantly cutaneous and included, leading to a total of 254 cases included for subsequent analysis. All cases were re-reviewed by a board-certified pathologist for inclusion in the present cohort.”

The sequencing protocol and methods are not outlined.

We apologize for not including this information in our initial submission. Sequencing was performed as per standard of care using the UCSF500 panel as has been described previously in multiple studies (PMID: 35584348, 32139107, 31024753, 35291666, 35321431). We have now added the following detail to our methods on page 8: “DNA sequencing was performed at the UCSF Clinical Cancer Genomics Laboratory (CCGL) using the UCSF500 (https://genomics.ucsf.edu/UCSF500) CLIA certified targeted DNA next generation sequencing assay obtained as part of routine clinical care. Briefly, this assay uses a custom bait library (Roche Nimblegen) to cover the genomic sequence of 529 cancer related genes and select introns of 47 genes. High throughput sequencing of captured libraries is performed using the Illumina NovaSeq6000, aligned to the human reference genome, and variants are called using an internal pipeline then filtered before undergoing manual review to assign function based on known pathogenic alterations and predicted effects on protein by a team of board-certified pathologists and the UCSF CCGL team.”

How was MSI status established?

We again apologize for omitting these details from the initial submission. MSI was calculated based on the percentage of tested loci that demonstrated instability using the MSIsensor (PMID 24371154) which is performed as part of UCSF500 genetic testing profiled during routine clinical care.  A cutoff of 30% or greater was used to classify tumors as “MSI-high” as is done per standard of care genomic testing with the UCSF500 panel.  Given the highest percentage of microsatellite instability in cohort was 4.7%, none of the cases were deemed MSI-high. We have added this information to the methods on page 8 as follows: “

Penetrance (ratio of wt vs. mutated reads) of driver mutations and co-mutations should be analyzed.

This is an excellent point that we spent time considering when preparing this manuscript. We initially included an analysis of variant allele frequency, particularly with driver genes, to understand how this might impact outcomes or tumor phenotype.  However, after discussion with our pathology team, this appeared to be significantly confounded by tumor cell content, which can be highly variable between pathologic specimens, and moreover, many of our reports did not contain information regarding tumor cell percentage in the specimen.  Given melanoma specimens are often quite small (from skin biopsies or as part of MOHS resections) and contain surrounding normal tissue, we felt it was difficult to accurately estimate the ratio of wild type versus mutated reads in the sample without having confounding from the surrounding normal tissue.  For this reason, we decided to exclude this analysis from the present manuscript although we are working on approaches to mitigate this limitation, which we hope to be the subject of future studies.

Tumorigenic driver genes can also be hit by bystander mutations. The pathogenic potential of mutations in driver genes should be analyzed in the co-mutation group.

We apologize for not better describing our targeted sequencing approach in the initial submission. Mutations were only included in the analysis if they were deemed to be pathogenic or likely pathogenic based on review by our UCSF Clinical Cancer Genomics Laboratory (CCGL) team overseen by a board-certified pathologist with expertise in molecular alterations in cancer. As noted in response to the comment above requesting details on sequencing methods, we have now added this information to the methods.

The mutual exclusivity of mutations of the three main tumor suppressor genes and features of tumors with double TSG hits should be analyzed.  TSG mutations and therapy response should be analyzed.

This is a great point and idea. The three main tumor suppressor genes found in our study were TERT, CKDN2A/B and TP53.  None of the tumor suppressor genes were found to be mutually exclusive with one another in our analysis.  There were 121 cases with mutations in 2 of 3 most common tumor suppressor genes and an additional 23 cases with mutations identified in all 3 of the most common tumor suppressor genes.  We have detailed this information in the results of the revised manuscript on  page 11-12 “The three most commonly altered tumor suppressor genes included TERT, CDKN2A/B and TP53.  A total of 121 cases demonstrated mutations in 2 of these 3 genes and an additional 23 cases were “triple hit” in terms of tumor suppressor genes with deleterious mutations in TERT, CDKN2A/B and TP53.” and on page 15 “There was a trend towards improved progression free survival for patients with mutations in at least 2 of the 3 most commonly mutated tumor suppressor genes (TERT, CKDN2A/B, and TP53) with a HR for progression of 0.613 (95% CI 0.373 – 1.009, p = 0.054), although this did not reach statistical significance.”

The frequency of MAP-kinase pathway mutations found in TWT cases should be analyzed for BRAF, NRAS and NF1 cases.

This is yet another excellent point.  The most frequent MAP-kinase pathway mutations found in TWT cases included MAP2K1 (n=7 or 21.2%), SPRED1 (n=4 or 12.1%), KIT (n=3 or 9.1%) and FGF4/19 (n=13 or 9.1%).  In tumors with a BRAF, NRAS, or NF1 alteration (non TWT tumors), MAP2K1 mutations were identified in 7 of 221 (or 3.2%), SPRED1 mutations were found in 5 of 221 (or 2.3%), KIT in 7 of 221 (or 3.2%) and FGF4/19 in 10 of 221 (or 4.5%) of cases. We have added this information to the main text on page 14 as follows: “In tumors harboring a BRAF, NRAS, or NF1 mutation (non TWT tumors), MAP2K1 mutations were identified in 7 of 221 (or 3.2%), SPRED1 mutations were found in 5 of 221 (or 2.3%), KIT in 7 of 221 (or 3.2%) and FGF4/19 in 10 of 221 (or 4.5%).”

Round 2

Reviewer 2 Report

Comments and Suggestions for Authors

Thank you  for  the revision provided

Author Response

Thank you  for  the revision provided

We appreciate the positive evaluation of our revised manuscript

Reviewer 3 Report

Comments and Suggestions for Authors

The authors adequately replied to the issues raised. I have, however, one doubt about TERT that is now indicated as a tumor suppressor, which I think it is not. TERT promoter mutations present in over 70% of melanomas determine overexpression of TERT that contributes to telomer elongation needed for unlimited cell division numbers. Hence, it is considered to possess oncogenic functions. I am missing PTEN: did the authors not find any PTEN mutations? If so, information on its combination with CDKN2A and TP53 mutations wuld be interesting. 

Author Response

The authors adequately replied to the issues raised.

Thank you for the positive appraisal of our response.

I have, however, one doubt about TERT that is now indicated as a tumor suppressor, which I think it is not. TERT promoter mutations present in over 70% of melanomas determine overexpression of TERT that contributes to telomer elongation needed for unlimited cell division numbers. Hence, it is considered to possess oncogenic functions.

This is an excellent point and we apologize for this miscateogorization, as we meant to state these were the three most commonly co-mutated genes. We have amended the main text to now explicitly state the TERT alterations are an activating mutation in the promoter on page 6 of the revised text as follows:

"The three most commonly co-altered genes included activating TERT promoter mutations and loss of the tumor suppressors CDKN2A/B and TP53.  A total of 121 cases demonstrated mutations in 2 of these 3 genes and an additional 23 cases were “triple hit” in terms of alterations in TERT, CDKN2A/B and TP53."

I am missing PTEN: did the authors not find any PTEN mutations? If so, information on its combination with CDKN2A and TP53 mutations wuld be interesting. 

Thank you for raising this point - as stated on page 6 and shown in Figure 1a-b, a total of 25 tumors harbored PTEN alteration (typically deletion). Of these 25 events, 12 co-occurred with CDKN2A/B homozygous mutation and 10 co-occurred with TP53 mutation. We have now added this to page 6 of the main text as follows:

"In addition, there were 25 cases harboring PTEN alteration, of which 12 co-occurred with CDKN2A/B loss and 10 co-occurred with TP53 mutation."